# Stereoselective diversification of α-amino acids enabled by N-heterocyclic carbene catalysis

Hong Zhang[1,3], Yuxing Cai[2,3], Yuqi Fang[1,2,3], Yong Huang [2] ✉ & Jiean Chen [1] ✉

Chiral α-amino acids (AAs), essential to biological systems and drug design, drive demand for precise synthetic methods to access unnatural variants (UAAs) and stereochemically defined peptides. We report an N-heterocyclic carbene (NHC)-catalyzed strategy enabling enantioselective synthesis of α-(U)AA esters and peptides. Leveraging NHC-generated acyl azolium intermediates, this approach achieves dynamic kinetic resolution of racemic or chiral α-(U)AAs with broad substrate scope, including sterically hindered and unsaturated derivatives. Stereodivergent synthesis is accomplished via NHC-mediated proton shuttling, which usually furnishes enantio-complementary α-(U)AAs and peptides with >90% ee (de). Mechanistic studies establish that N,N'-diisopropylcarbodiimide activates α-(U)AAs to form oxazolone intermediates, which undergo NHC-mediated conversion to acyl azolium species. Divergent nucleophilic pathways are governed by chiral matching between catalyst and substrate, as evidenced by density functional theory (DFT) calculations revealing π-π interactions and steric effects as stereoselectivity determinants. The methodology's utility is also demonstrated in solid-phase peptide synthesis, achieving direct chirality transfer from racemic precursors to peptides with minimal epimerization. This work provides a catalytic platform for stereocontrolled α-(U)AA and peptide synthesis, with implications for chemical biology and peptide therapeutic development.

α-Amino acids (AAs), encompassing both natural and unnatural variants (UAAs), serve as fundamental building blocks for protein structure and function, underpinning biological systems and therapeutic design[1–9]. Recent advances in pharmaceutical chemistry and biotechnology have amplified the demand for enantiopure α-AAs, particularly in drug development, spurring innovations in asymmetric synthesis (Fig. 1A)[10–14]. Current approaches to access chiral α-AAs include isolation from protein hydrolysates, biocatalysis, and chemical synthesis[15–18]. While enzymatic methods, such as amino acid racemase-mediated interconversion of L- and D-AAs, offer step efficiency, their utility for α-UAAs is constrained by enzyme stability, catalytic activity,

and narrow substrate scope (Fig. 1B)[19–22]. In contrast, chemical synthesis has evolved as a versatile and scalable platform for constructing chiral α-(U)AAs. State-of-the-art methodologies, including dynamic kinetic resolution (DKR) of azlactones or carboxylic acids[23–27], N–H insertion reactions[28–32], and asymmetric hydrogenation[33,34], have significantly expanded the synthetic repertoire. These strategies further enable the preparation of peptides incorporating chiral α-UAAs, which show significant potential in the design of peptide-mimetic therapeutics[35,36]. However, most existing catalytic systems necessitate substrates with specific structural features or strategic premodification to align with their activation modes. Consequently,

[1]Pingshan Translational Medicine Center, Shenzhen Bay Laboratory, Shenzhen, China. [2]Department of Chemistry, The Hong Kong University of Science and Technology, Clear Water Bay, Kowloon, Hong Kong SAR, China. [3]These authors contributed equally: Hong Zhang, Yuxing Cai, Yuqi Fang.
✉ e-mail: yonghuang@ust.hk; chenja@szbl.ac.cn

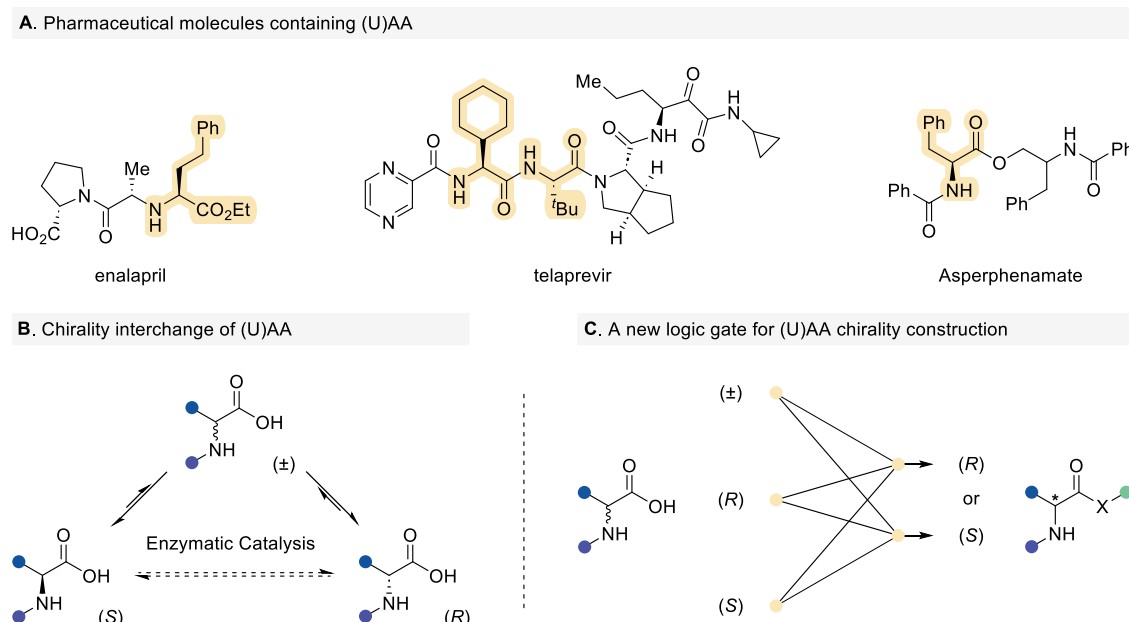

**Fig. 1 | Strategies for the transformation of α-(U)AAs. A** Selected examples of drug molecules containing amino acid modules. **B** Enzyme-catalyzed chirality interchange of an (unnatural) amino acid stereocenter. **C** Chiral amino acid construction enabled by the oriented transformation of (racemic) stereocenters.

developing innovative methodologies to access structurally diverse α-(U)AAs remains a critical challenge. Herein, we present a catalytic strategy that employs readily available racemic (U)AAs or enantiopure α-(U)AAs (L/D-configuration) as starting materials, enabling stereoselective conversion to target (U)AAs with precise stereochemical control. This approach leverages N-heterocyclic carbene (NHC) catalysis to bypass traditional substrate limitations, offering a streamlined route to chiral α-(U)AAs with tailored functional and stereochemical diversity (Fig. 1C).

Building on established principles of N-heterocyclic carbene (NHC) catalysis, we hypothesize that NHC-derived acyl azolium intermediates enable both the construction and stereochemical preservation of α-stereocenters in amino acids[37–42]. Upon activating carbonyl substrates—including anhydrides, imides, esters, amides, or aldehydes—the NHC catalyst engages the carbonyl carbon, facilitating chiral induction at the α-position via a proton shuttle mechanism[38,39,43–48]. This DKR process efficiently transfers stereochemical information from the NHC scaffold to the nascent α-stereocenter, aligning with our strategy for directed chiral α-AA synthesis (Fig. 2A). Herein, we introduce a methodology for chiral α-AA synthesis: sterically demanding chiral NHC catalysts drive enantioselective assembly of α-AA derivatives, while minimally hindered NHC variants preserve stereochemical fidelity during subsequent peptide elongation. This dual catalytic system resolves the competing demands of stereoselective synthesis and configurational stability, providing a unified platform for chiral α-AA construction and peptide stereocontrol (Fig. 2B).

## Results

### NHC-catalyzed stereoselective formation of α-amino esters

To test our hypothesis, we systematically explored the stereoselective protonation of α-amino acids (**1**) using diphenylmethanol (**2a**) as a sterically demanding nucleophile. Initial screening identified *N,N′*-diisopropylcarbodiimide (DIC) as the optimal activating agent, demonstrating superior efficiency among various coupling reagents. Further optimization revealed that combining 20 mol% NHC precatalyst **3a** with 1 equivalent Cs$_2$CO$_3$ in dichloromethane (0.05 M) efficiently promoted the formation of target α-amino ester **4**,

achieving excellent yield and enantioselectivity (see Supplementary Tables S1–S3).

With optimized reaction conditions established, we evaluated the substrate generality of this transformation (Fig. 3). The methodology demonstrated broad compatibility with both natural and unnatural amino acids. Racemic N-terminal-protected amino acids underwent convergent stereoselective conversion to (*S*)-configured derivatives (Phe, Trp, Ala, Met; **4a-4d**, >90% ee average). α-Benzyl-substituted substrates, featuring electron-rich and electron-deficient aryl groups, proceeded smoothly with high enantioselectivity (products **4e-4j**, 90–97% ee). A thiophene-containing phenylalanine analog was efficiently synthesized under these conditions, requiring only a single additional step to generate the racemic precursor from α-bromo carboxylic acid (**4k**, 87% yield, 94% ee; see SI). Notably, steric hindrance from α-aryl substituent did not impede the reaction, enabling the synthesis of chiral α-aryl analog **4l** (65% yield, 91% ee). α-Alkyl-substituted substrates underwent DKR with high efficiency, yielding target chiral UAAs (**4m-4p**, 92–97% ee). Crucially, this strategy exhibited superior functional group tolerance compared to conventional transition metal- or enzyme-catalyzed approaches, preserving unsaturated side chains (**4q-4r**) without double/triple bond degradation—a critical advantage for post-synthetic modifications. The methodology further accommodated secondary α-cyclohexyl substituents (**4s**, 82% yield, 90% ee), though tertiary α-substituents completely suppressed reactivity (see Supplementary Fig. S5).

Using phenylalanine (Phe) as a model substrate, we systematically investigated the influence of N-protecting groups (PGs) on the reaction (Fig. 4). Substituted benzoyl derivatives proved highly effective (see Supplementary Fig. S3), while other acyl PGs—including acetyl, heterocyclic, cinnamyl, alkyl, and amide groups—were also well tolerated (**4ab-4af**). This diversity in compatible N-terminal PGs expands the range of feasible deprotection strategies for subsequent peptide modification, enhancing adaptability across different applications. We next examined nucleophiles for catalyst turnover, a critical component of the acyl-azolium catalytic cycle. Primary and secondary alcohols, including functionalized variants (e.g., hydroxyproline derivatives), performed exceptionally well (**4ag-4ak**). We extended our investigation to peptide substrates with this demonstrated functional group

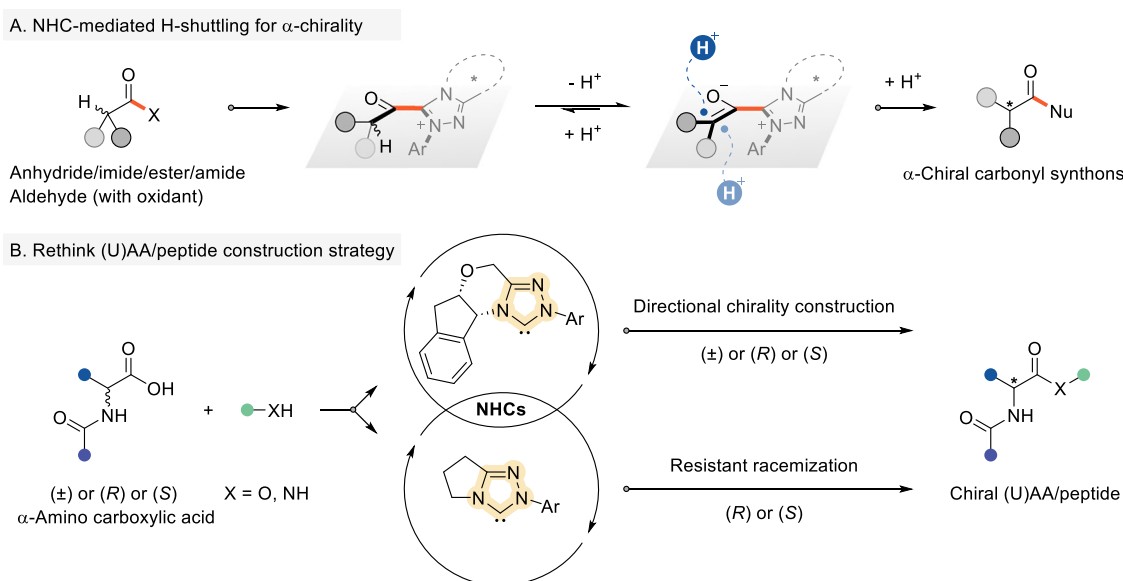

**Fig. 2 | NHC catalyzed the construction of α-(U)AAs. A** NHC-mediated H-shuttling for constructing carbonyl α-stereocenters. **B** NHC-mediated approach enabling the synthesis of chiral (U)AAs and peptides (this study).

tolerance. Under standard conditions, racemic N-terminal peptide-linked amino acids underwent asymmetric proton shuttle via DKR, affording chiral peptide products with excellent stereocontrol (**6a**-**6g**). The protocol retained efficiency even with UAA moieties embedded in the peptide backbone, yielding chiral peptides with diverse side chains (**6h**-**6j**). The successful DKR of peptides into their chiral ester derivatives highlights the potential of this methodology for peptide chemistry and related applications.

Control experiments were performed to elucidate the reaction mechanism (see Fig. 5 and Supplementary Fig. S13). The transformation required all key components (NHC catalyst, base, and DIC) for optimal performance. Notably, oxazolone intermediate **Int-1a** was identified as a crucial species formed without NHC catalyst. Screening of coupling reagents revealed EDCI provided a high yield (97%), albeit with reduced enantioselectivity (86% ee), while HATU and CDI proved ineffective, yielding only trace products. Isolation and subsequent treatment of **Int-1a** under optimized conditions (without DIC) afforded product **4a** in high yield with excellent enantioselectivity, confirming oxazolone's role as a key intermediate in the enantioselective transformation, as well as activated p-nitrophenyl ester has similar reactivity (see Fig. 5A and Supplementary Fig. S4). Further evaluation of N-PGs demonstrated that while Cbz, Fmoc, and Boc groups provided high yields of amino esters, they exhibited poor stereocontrol. This distinction arises from DIC's ability to promote oxazolone formation with Bz and Ac groups but not with Cbz, Fmoc, or Boc as PGs (Fig. 5B). The methodology proved effective for α-chirality inter-conversion between (*S*)- and (*R*)-configurations (Fig. 5C). Starting from either (*S*)-, (*R*)-, or racemic **1ab**, a single stereoisomer (*R*)-**4ab** was obtained. Using *ent*-NHC **3a** as a catalyst inverted the selectivity to furnish (*S*)-**4ab**. This stereochemical control was further validated using (*rac*)-**1aa** as the substrate and N-fmoc-*trans*-4-hydroxy-L-pro-line methyl ester as the nucleophile, which enabled the synthesis of a single stereoisomer (*S*) or (*R*)-**4aj**. These results collectively support a mechanism involving initial substrate racemization followed by NHC-mediated α-chiral reconstruction.

**NHC-catalyzed racemization-free peptide synthesis application**
Building on these insights, we systematically evaluated the methodology's capacity to stabilize α-stereocenters and suppress

racemization during peptide synthesis (see Fig. 6A and Supplementary Table S4)[49,50]. Employing amino acid hydrochloride **2b** as a nucleophile with minimally hindered NHC catalyst **7a** could effectively minimize racemization. The reaction efficiently operated under base-free conditions, outperforming direct coupling approaches that exhibited moderate yield and excellent stereoselectivity (73% yield, 99% es vs. entries 1–3). Introducing exogenous bases significantly exacerbated racemization, underscoring the importance of base-free circumstances (entry 4, 34% yield, 29% es). Heat-map analysis of N-PGs revealed enhanced yields and diastereomeric excess in the presence of **7a**, confirming its role in racemization suppression (products **8b**–**8f**). We further designed a tandem sequence integrating chiral (U)AA synthesis and peptide elongation, enabling direct access to enantiopure peptides from racemic N-protected amino acids (product **8a**, 62% yield, 94% de). Expanding this concept, we developed a Wang resin-based solid-phase peptide synthesis (SPPS) protocol that simultaneously establishes α-chirality and peptide bond, delivering target peptides with high stereo fidelity (product **8c**, 20% yield, 97% de). Further scale-up reaction and representative examples proved highly effective (see Supplementary Figs. S7, S8).

## DFT calculation
Density functional theory (DFT) calculations were performed to elucidate the stereochemical control governing chiral reconstruction (see Fig. 7 and Supplementary Figs. S14–16). The computational studies delineate the stereoselective transformation pathway, beginning with NHC-catalyzed oxazolone ring-opening. Transition state analysis reveals a 1.2 kcal/mol energy preference for (*S*)-**TS₁** over (*R*)-**TS₁**, driven by stabilizing phenyl π-π interactions between the substrate and NHC catalyst. Subsequent CsHCO₃-mediated pro-tonation generates acyl azolium intermediate **Int₂**, where steric constraints render the *R*-configured species 2.9 kcal/mol higher in energy than its *S*-counterpart. Two divergent pathways emerge from **Int₂**: (1) direct CsHCO₃-assisted alcohol addition via **TS₃**, or (2) enolate formation (**Int-Z/Int-E**) followed by alcohol addition through **TS₅**. The former pathway dominates, with (*S*)-**TS₃** exhibiting a 2.5 kcal/mol energy advantage over (*R*)-**TS₃** due to favorable substrate-alcohol π-π interactions (see Supplementary Fig. S17). This step, characterized by an activation barrier of 17.5 kcal/mol,

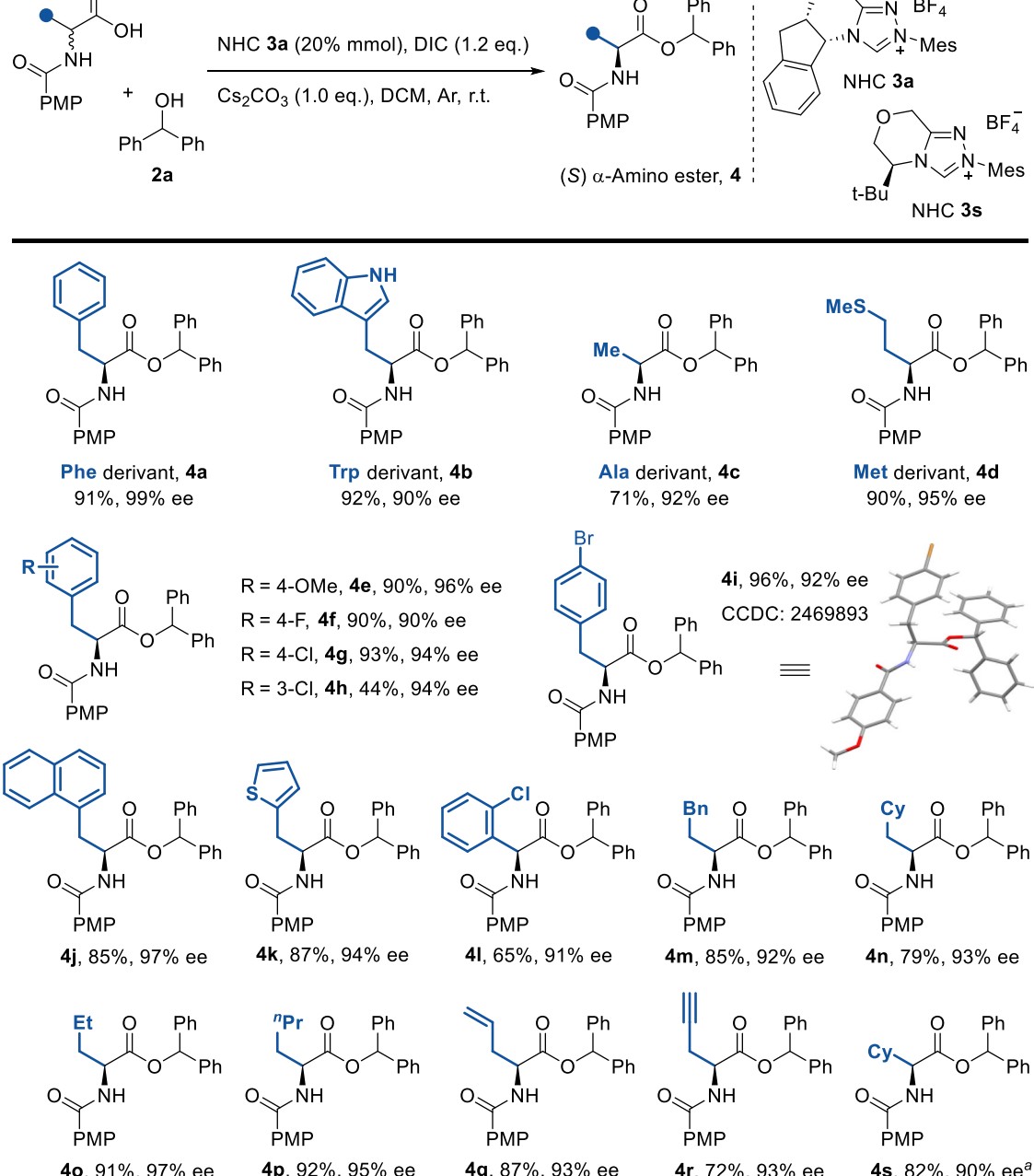

**Fig. 3 | The scope of chiral α-amino acids.** Reaction conditions: A mixture of α-AAs 1 (0.05 mmol), diphenylmethanol 2a (0.075 mmol), NHC-3a (20 mol%), Cs₂CO₃ (0.05 mmol) and DIC (0.06 mmol) dissolved in DCM (0.05 M) was stirred at r.t. for 2–4 h. Isolated yields are presented. Chiral HPLC determined ee values. a NHC-**3s** was applied instead of NHC-**3a**.

constitutes the rate-determining step. In contrast, the enolate pathway is kinetically disfavored, requiring significantly higher activation energy (($S$)-**TS₅** = 32.7 kcal/mol). These findings collectively establish that stereoselectivity arises from NHC-mediated modulation of kinetic (transition state stabilization) and thermodynamic (intermediate stabilization) factors, with the catalyst dictating the energy landscape and stereochemical trajectory.

## Discussion

We present an NHC-catalyzed strategy for the stereocontrolled synthesis of chiral α-amino acids, enabling direct access to both natural and unnatural variants with precise stereochemical fidelity. This methodology unifies chiral α-(U)AA construction and peptide synthesis into a single catalytic framework, resolving persistent challenges

in traditional approaches, including racemization and configurational instability. By leveraging NHC catalysts, the platform achieves dynamic stereochemical interconversion between S- and R-configured α-(U)AAs while maintaining chirality during peptide elongation—a dual capability critical for designing bioactive molecules. Operationally simple, base-free conditions and broad functional group tolerance accommodate diverse substrates, from sterically hindered α-aryl groups to unsaturated side chains, without compromising stereoselectivity. Mechanistic studies reveal that stereochemical outcomes are governed by NHC-mediated modulation of the reaction energy landscape, where π-π interactions and steric effects collaboratively dictate transition state preferences. This work bridges synthetic precision and biomolecular engineering, offering a versatile platform for biomolecular design and the development of peptide therapeutics.

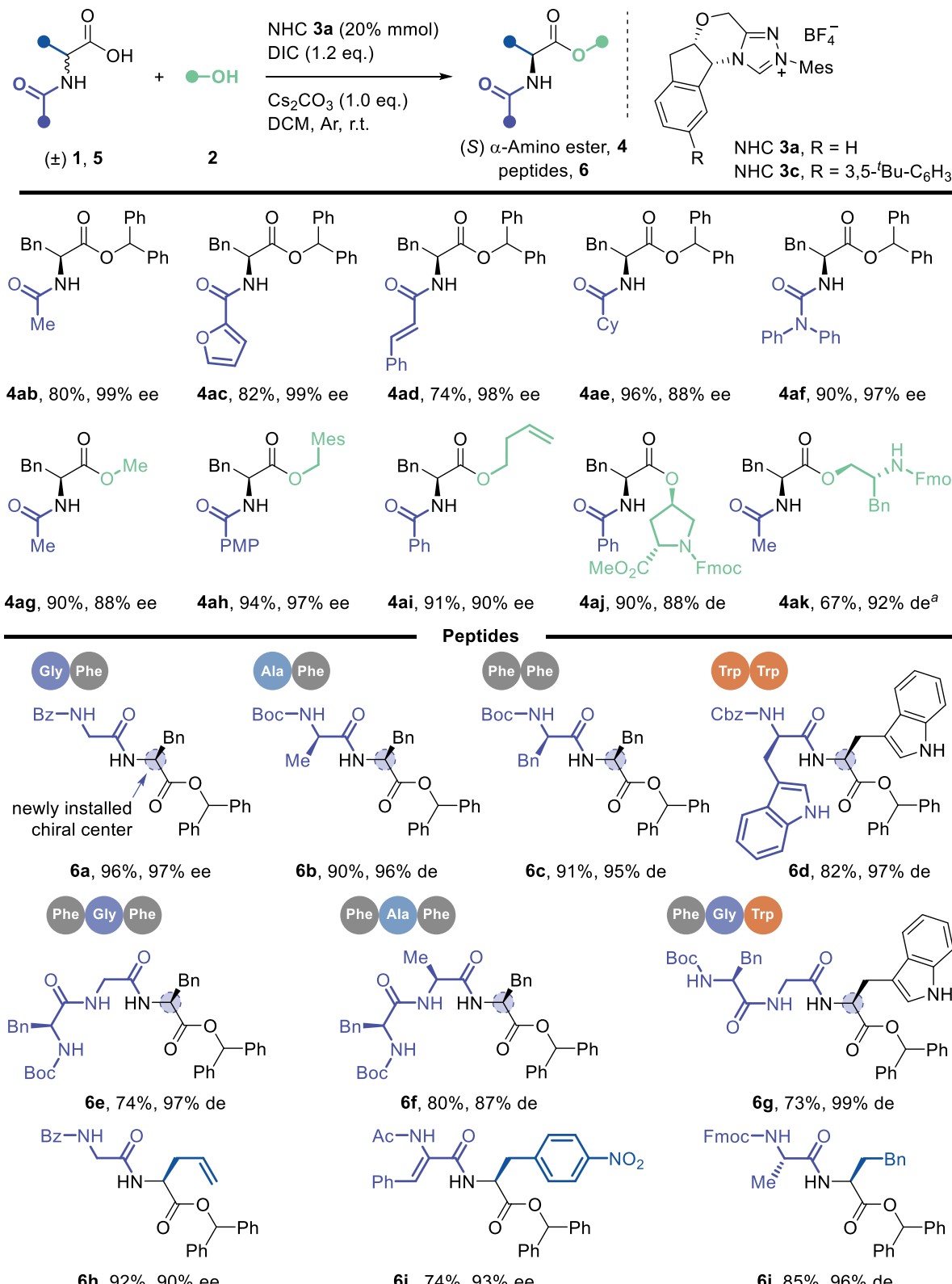

**Fig. 4 | The scope of substrates with different N-PGs and peptides.** Reaction conditions: A mixture of α-AAs **1** or peptides **5** (0.05 mmol), alcohol (0.075 mmol), NHC-**3a** (20 mol%), Cs$_2$CO$_3$ (0.05 mmol) and DIC (0.06 mmol) dissolved in DCM (0.05 M) was stirred at r.t. for 4–10 h. Isolated yields are presented. Chiral HPLC determined ee or de values. a NHC-**3c** was applied instead of NHC-**3a**.

**A**. Control experiment

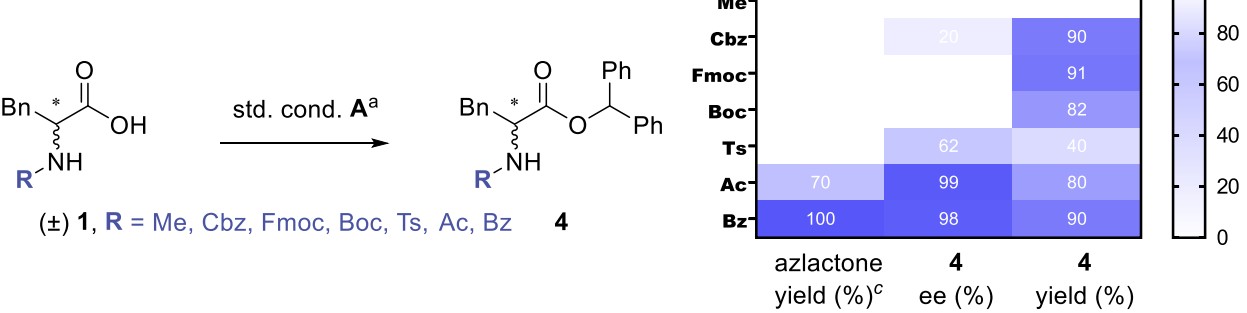

| Entry | differ from std. con. **A** | yield of **4a** (%) | ee (%) |
|-------|------------------------------|---------------------|--------|
| 1 | no NHC or base or DIC | 0(ND) | -- |
| 2 | EDCI instead of DIC | 97 | 86 |
| 3 | HATU or CDI instead of DIC | <10 | -- |
| 4[b] | azlactone **Int-1a** as substrate | 95 | 99 |

azlactone **Int-1a**
w/o NHC, 97% yield

**B**. N$^\alpha$-protecting group impact

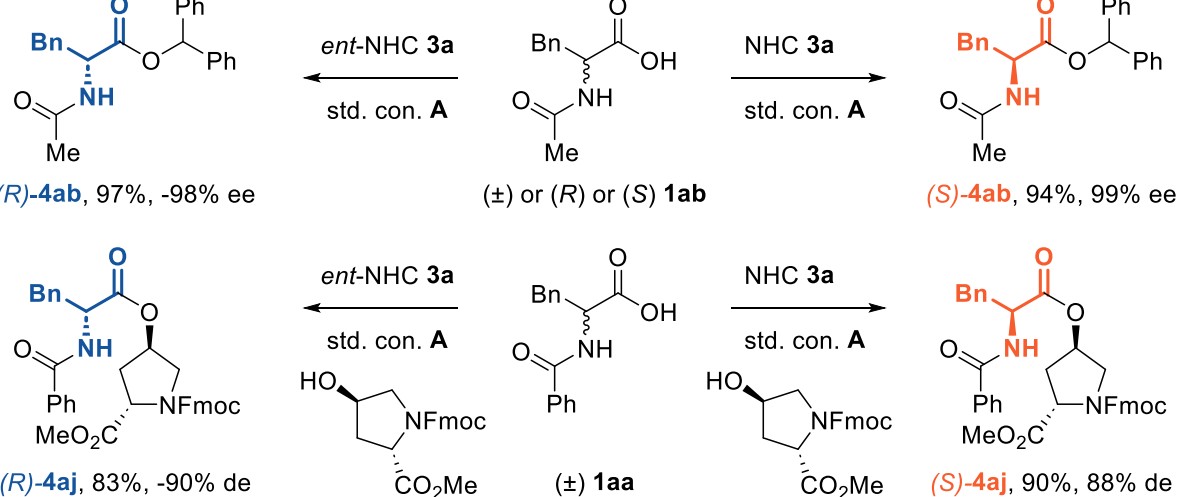

**C**. Chiralty reconstruction

*(R)*-**4ab**, 97%, -98% ee          (±) or (*R*) or (*S*) **1ab**          *(S)*-**4ab**, 94%, 99% ee

*(R)*-**4aj**, 83%, -90% de          (±) **1aa**          *(S)*-**4aj**, 90%, 88% de

**Fig. 5 | Mechanistic study about NHC-catalyzed esterification of AAs. A** Standard conditions$^a$ for control experiment: A mixture of AAs **1a** (0.05 mmol), alcohol **2a** (0.075 mmol), NHC-**3a** (20 mol%), Cs$_2$CO$_3$ (0.05 mmol) and DIC (0.06 mmol) dissolved in DCM (0.05 M) was stirred at r.t. for 2–10 h. Isolated yields are presented. Chiral HPLC determined ee and de values, see Supplementary Fig. S2. $^b$Oxazolone **Int-1a** as starting material without DIC in the std. cond. A, see Supplementary Fig. S1. **B** N$^a$-protecting group impact. $^c$Direct synthesis of oxazolone under DIC. **C** α-chirality interconversion between *(S)-* and *(R)-*configurations.

**A.** Control experiment

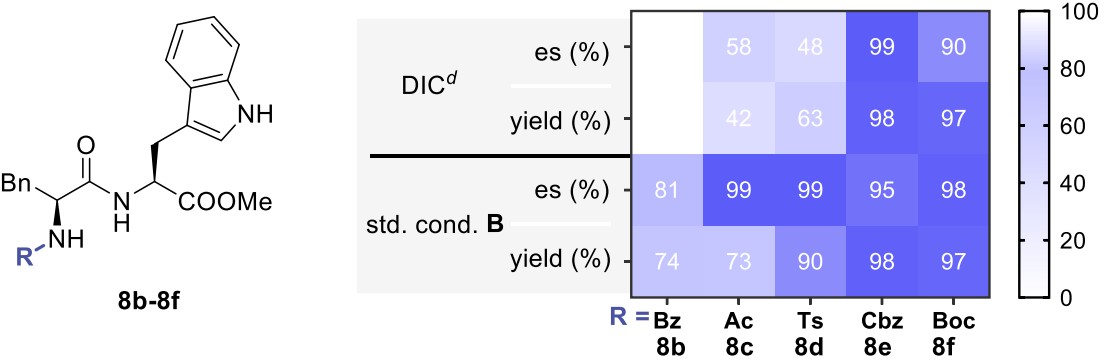

| entry | conditions[b] | yield of **8c** (%) | es (%) |
|---|---|---|---|
| 1 | DIC (1.2 eq.) | 42 | 64 |
| 2 | DIC (1.2 eq.), HOBt (1.2 eq.) | 70 | 75 |
| 3 | HATU (1.5 eq.) DIPEA (3 eq.) | 71 | 90 |
| 4[c] | NHC **7a** (0.2 eq.), DIC (1.2 eq.), DIPEA (1.2 eq.) | 34 | 29 |

**8b-8f**

|  |  | Bz 8b | Ac 8c | Ts 8d | Cbz 8e | Boc 8f |
|---|---|---|---|---|---|---|
| DIC[d] | es (%) |  | 58 | 48 | 99 | 90 |
|  | yield (%) |  | 42 | 63 | 98 | 97 |
| std. cond. **B** | es (%) | 81 | 99 | 99 | 95 | 98 |
|  | yield (%) | 74 | 73 | 90 | 98 | 97 |

R = Bz **8b**, Ac **8c**, Ts **8d**, Cbz **8e**, Boc **8f**

**B.** Application

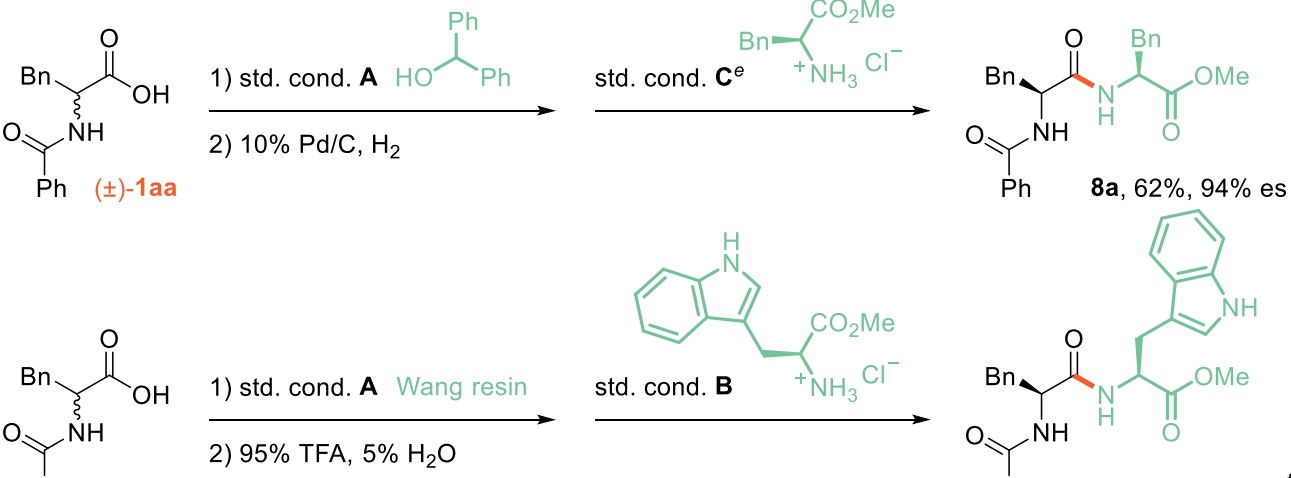

**Fig. 6 | NHC-catalyzed racemization-free peptide synthesis. A** Standard conditions[a] for control experiment: a mixture of chiral AAs **1ab** (0.05 mmol), amine **2b** (0.055 mmol), NHC-**7a** (20 mol%) and DIC (0.06 mmol) dissolved in DCM (0.05 M) was stirred at r.t. for 3–6 h. Isolated yields are presented. Chiral HPLC determined de and es values, see Supplementary Fig. S6. [b]Compounds **1ab** and **2b** dissolved in DCM (0.05 M) were stirred at r.t. [c,d]For concrete operations, see Supplementary Figs. S9 and S10. **B** Application in peptide synthesis. [e]Std. cond. C: step 1: MYTsA (1.2 eq.) in DCM; step 2: H-L-Phe-OMe.HCl (1.1 eq.) in DMF, see Supplementary Fig. S11. [f]For concrete operations, see Supplementary Fig. S12.

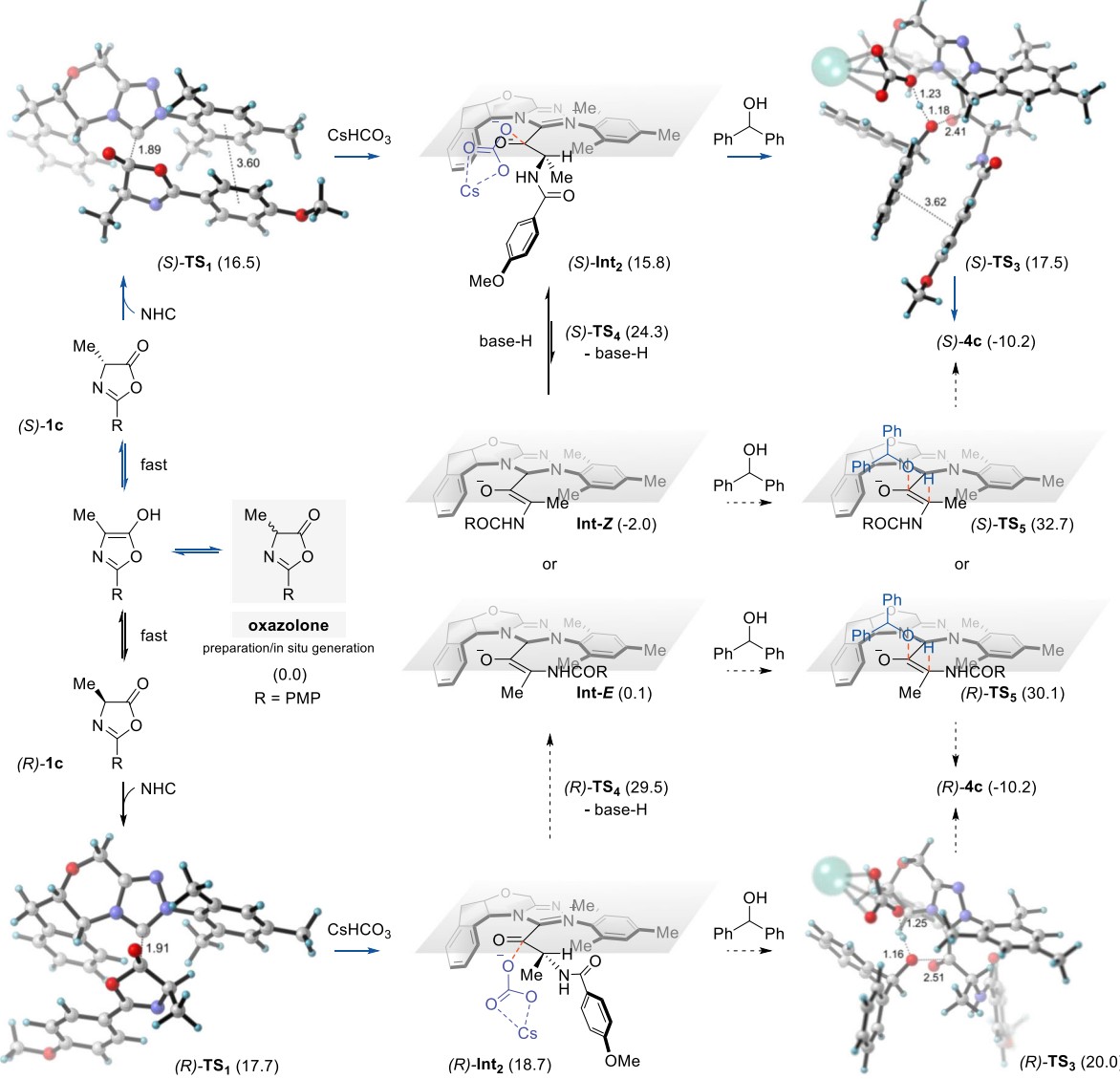

**Fig. 7 | Mechanistic investigation of NHC-catalyzed alcoholysis of oxazolone.** The relative Gibbs free energies are reported from the SMD(DCM)-B3LYP-D3(BJ)/ def2tzvp//B3LYP-D3(BJ)/def2svp level of theory, and the values are given in kcal/mol.

## Methods

### General method for the NHC-catalyzed esterification of AAs and peptides

In an oven-dried 5 mL Schlenk tube equipped with a magnetic stir bar, α-amino acids **1** or peptides **5** (0.05 mmol), NHC precatalyst **3a** (0.01 mmol, 20 mol%), alcohol **2** (0.075 mmol, 1.5 equiv), and Cs$_2$CO$_3$ (0.05 mmol, 1 equiv) were added, followed by 1 mL of dry dichloromethane (DCM). The resulting mixture was degassed and backfilled with argon (3 cycles), after which DIC (0.06 mmol) was added. The tube was sealed with a screw cap, and the reaction mixture was stirred vigorously at room temperature (25 °C) for 10 h. Upon completion (as monitored by TLC), the crude mixture was purified by flash column chromatography (petroleum ether/ethyl acetate = 10:1 to 4:1) to afford the corresponding ester products **4** or **6**.

### General method for the NHC-catalyzed racemization-free peptide synthesis

In a glove box under an argon atmosphere, the NHC precursor was pre-activated with NaH (1.2 equiv) for 1 h. After filtration, the resulting solution of free NHC-**7a** (0.01 mmol, 20 mol%) was added to a reaction

mixture containing α-amino acid **1** (0.05 mmol) and amine **2b** (0.055 mmol, 1.1 equiv) in an oven-dried 5 mL Schlenk tube equipped with a magnetic stir bar. Dry dichloromethane (DCM, 1 mL) and DIC (0.06 mmol) were subsequently added inside the glove box. The tube was sealed with a screw cap, and the mixture was stirred vigorously at room temperature (25 °C) for 6 h. Upon complete consumption of α-amino acid **1** (monitored by TLC), the crude reaction mixture was purified by flash column chromatography (petroleum ether/ethyl acetate = 2:1 to 1:1) to afford the desired peptide product **8**.

## Data availability

The X-ray structure data generated in this methodology have been deposited in the Cambridge Crystallographic Data Centre (CCDC 2469893 for **4e**, 2469924 for **4al**). Copies of the data can be obtained free of charge via https://www.ccdc.cam.ac.uk/structures/. Experimental procedures, characterizations of new compounds, and all other data supporting the findings are available in the Supplementary Information. The Cartesian coordinates of intermediates and transition states can be found in the Supplementary Data. Data supporting the findings of this manuscript are also available from the corresponding authors upon request.

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

## Acknowledgements

This work was financially supported by the Guangdong Basic and Applied Basic Research Foundation (2025B1515020070), Hong Kong RGC (16302122 and 16308224), HKUST-Kaisa Joint Research Institute Research Project (HKJRI-043) and the China Postdoctoral Science Foundation (2023M742419). We are grateful to the HPC core facility of the Shenzhen Bay Laboratory for their assistance with the DFT calculation.

## Author contributions

J.C. and Y.H. directed the project. H.Z. and Y.C. conducted the experiments. Q.F. performed the DFT calculation. All authors contributed to analyzing the experimental results and writing this manuscript.

## Competing interests

The authors delare no competing interests.
