## [Transparent Peer Review file · Nature Communications]

Stereoselective Diversification of α -Amino Acids Enabled by N-Heterocyclic Carbene Catalysis

Corresponding Author: Professor Jean Chen

Version 0:

Reviewer comments:

Reviewer #1

(Remarks to the Author)

This study presents a novel strategy for the chiral construction of α -(unnatural) amino acids within an N-heterocyclic carbene (NHC) catalytic system. The methodology demonstrates compatibility with both chiral and racemic N-protected α -amino acid substrates, with the catalyst's stereochemical information exerting complete control over the absolute configuration of the product. Furthermore, leveraging this catalytic approach, the authors developed a "base-free" peptide bond ligation strategy, showcasing preliminary compatibility with amino acid substrates and potential applicability in solid-phase peptide synthesis. Mechanistically, the work proposes oxazolone intermediates as critical species and highlights the involvement of the NHC catalyst in enabling dynamic kinetic resolution (DKR), diverging from the conventional "enolate-protonation" pathway mediated by NHC-bound acyl azolium intermediates. This represents a notable mechanistic insight. Given its integration of innovative catalytic mechanisms and practical applications, this work aligns well with the journal's focus on mechanistic depth and application-driven advances in catalysis. I recommend publication pending the resolution of the following minor revisions:

1. Previous NHC-catalyzed DKR alcoholysis reactions typically employ activated esters. As this work utilizes in situ-generated oxazolone intermediates, could the authors clarify whether pre-activated amino acid esters (e.g., p-nitrophenyl or pentafluorophenyl esters) were tested as substrates?
2. DIC was identified as the optimal activating reagent, while EDCI yielded inferior stereoselectivity despite structural similarity. What mechanistic rationale (e.g., steric, electronic, or kinetic factors) might explain this disparity in stereocontrol?
3. For Ts- and Cbz-protected substrates in Figure 5b, stereoselectivity was observed despite no oxazolone intermediate formation. Could this imply a transient active ester pathway (via DIC-activated amino acids) that generates acyl azolium intermediates directly upon NHC interaction, thereby enabling stereocontrol?
4. To enhance mechanistic clarity, including 3D structural data for all intermediates and transition states in the Supporting Information is recommended. Only selected structures (Figure 7, main text) are currently provided.
5. SI Table 4 reports limited diastereoselectivity (72% de) for amino acid ester substrates in chiral peptide synthesis. Were efforts made to optimize this transformation (e.g., activating reagents, catalysts, solvents, temperatures)? Additionally, is compatibility with sulfonamide-based reagents feasible?
6. Manuscript Corrections:
 - a) A discrepancy exists in the discussion of Int2 conformers: The cited 2.9 kcal/mol energy difference (Figure 7) conflicts with the textual description. Clarification is required.
 - b) In Figure 6a (heatmap), the yield and es values for the Ac-protected substrate under condition B appear mislabeled. Please verify and correct.

Reviewer #2

(Remarks to the Author)

This manuscript describes the stereoselective diversification of α -amino acids using N-heterocyclic carbene catalysts in the presence of a Cs_2CO_3 base. The resulting peptides exhibit high enantio- and diastereoselectivities. The synthetic methodology presented is significant for applications in organic synthesis and medicinal chemistry. They also used DFT calculations to gain insight into the mechanisms, and origin of the stereoselectivities. As a reviewer with expertise in DFT calculations, I would like to offer several comments regarding the computational aspects of the paper.

1. In Figures 3–5, the authors use the term "enantoexcess (ee)," whereas elsewhere they refer to "enantioselectivity (es)." Although these terms are related, the inconsistency may confuse readers. The authors are encouraged to standardize the terminology throughout the manuscript.

2. Dispersion Correction Clarification: In the caption of Figure 7, the authors mention the use of D3 dispersion corrections with Becke–Johnson (BJ) damping. However, the Supplementary Information does not specify whether BJ damping was applied (maybe zero damping?). The authors should clarify whether B3LYP-D3(BJ) or B3LYP-D3 was used in their calculations.

3. Mechanistic Insight: Figures 7 and S1 indicate that the (S)-pathway is favored over the (R)-pathway. However, the explanation focuses solely on steric repulsions in the acyl azolium intermediate (Figure S3). The authors are encouraged to perform energy decomposition analyses (EDA) and noncovalent interaction (NCI) analyses on the transition states to deepen the mechanistic understanding. These approaches could provide more detailed insights into the origin of stereoselectivity.

Additional Comments:

4. In Figure S1, "Ts" should be corrected to "TS" to maintain consistency with standard nomenclature.

5. In the computational methods section of the Supplementary Information, the authors should specify how Gibbs free energies were calculated, including the temperature and pressure conditions used.

Reviewer #3

(Remarks to the Author)

In the manuscript by Jiean Chen, Yong Huang et al., an N-heterocyclic carbene (NHC)-catalyzed strategy enabling enantioselective synthesis of α -(U)AA esters and peptides is reported. Stereodivergent synthesis of both enantiomers of chiral α -Aas derivatives was accomplished with >90% ee (de) via the DIC-initiated formation of racemic azlactones followed by efficient deracemization by means of NHC-generated acyl azolium intermediates and proton shuttling. The reaction mechanism was supported by control experiments and DFT calculations. The method is applicable to diverse substrates and demonstrates a broad functional group tolerance. It may be useful as a potential platform for asymmetric synthesis of peptide medications.

However, manuscript has some drawbacks which require major corrections.

- There is no data on scalability of the proposed NHC-based deracemization procedure which is crucial for biomolecular design and the development of peptide therapeutics.
- Experimental details of the declared Wang resin-based solid-phase peptide synthesis (SPPS) protocol should be added and discussed.
- The CCDC number «2223535» provided by the authors in the manuscript (Figure 3) and the ESI (Section «determination of absolute configuration», Page 82) must be checked and corrected. Quite different compound with the same CCDC number has been recently reported by another research team (Zi-Chao Wang, Xiaohua Luo, Jia-Wen Zhang, Chen-Fei Liu, Ming Joo Koh, Shi-Liang Shi, Nature Catalysis, 2023, 6, 1087, DOI: 10.1038/s41929-023-01037-9).
- One question more. The authors determined the absolute (S)-configuration of product 4e basing on the X-ray crystal structure. However, the determined space group «P -1» is centrosymmetric in which only racemic or achiral substances can crystallize. It is not clear, how the authors could determine the absolute configuration using the P-1 crystal.
- Melting points of solid products 4, 6 and 8 are absent in the ESI.
- Structural formulas of NHC-3s and NHC-3c mentioned in captions to Figures 3 and 4 should be depicted not only in the SI but also in the manuscript.
- I recommend the authors to measure yield, ee and dr values in percents in rectangles embedded to Figures 5b and 6a.
- Figure 5c. The hydroxyproline ester 4aj can't be obtained under standard condition A including usage of nucleophile 2a.
- Page 7, line 166: The 99% es – is it a "moderate" stereoselectivity?
- Pages in the ESI should be numerated.

The manuscript may become publishable if the reviewer comments are fully addressed.

Reviewer #4

(Remarks to the Author)

Version 1:

Reviewer comments:

Reviewer #1

(Remarks to the Author)

I appreciate the authors' effort to improve the manuscript and the authors have addressed all my concerns.

Reviewer #2

(Remarks to the Author)

The authors have revised the manuscript in response to the reviewers' comments. This reviewer concurs with the authors' responses and revisions, and recommends the manuscript for publication without further changes.

Reviewer #3

(Remarks to the Author)

The authors revised the original manuscript thoroughly according to the review comments. However, one more question arose after the revision. In Supplementary Information, page 85 there is no scheme 12, which is expected illustrate solid phase synthesis of compound 8c in detail. Step 1 also raises questions as it describes synthesis of "desired product" 1ab from the same compound 1ab. The necessity of this step and a role of the Wang resin in this "transformation" is not clear. I recommend publication of this manuscript after the authors address this comment.

Reviewer #4

(Remarks to the Author)

We would like to sincerely thank the reviewers, for the efficient work and instructive suggestions on "*Stereoselective Diversification of α -Amino Acids Enabled by N-Heterocyclic Carbene Catalysis*" (Manuscript ID: NCOMMS-25-29470-T). We have carefully revised this manuscript according to the reviewers' recommendations, and the detailed changes and arguments are highlighted. The follow-up is the point-to-point response to the reviewer's comments.

Reviewers' Comments to Author (Responses are highlighted in blue):

Reviewer #1:

This study presents a novel strategy for the chiral construction of α -(unnatural) amino acids within an N-heterocyclic carbene (NHC) catalytic system. The methodology demonstrates compatibility with both chiral and racemic N-protected α -amino acid substrates, with the catalyst's stereochemical information exerting complete control over the absolute configuration of the product. Furthermore, leveraging this catalytic approach, the authors developed a "base-free" peptide bond ligation strategy, showcasing preliminary compatibility with amino acid substrates and potential applicability in solid-phase peptide synthesis. Mechanistically, the work proposes oxazolone intermediates as critical species and highlights the involvement of the NHC catalyst in enabling dynamic kinetic resolution (DKR), diverging from the conventional "enolate-protonation" pathway mediated by NHC-bound acyl azolium intermediates. This represents a notable mechanistic insight. Given its integration of innovative catalytic mechanisms and practical applications, this work aligns well with the journal's focus on mechanistic depth and application-driven advances in catalysis. I recommend publication pending the resolution of the following minor revisions:

Our response: We would like to thank Reviewer 1 for his/her recognition of the overall research work and the evaluation of innovation. We have carefully considered and addressed the concerns raised, as detailed below.

Comments:

1. Previous NHC-catalyzed DKR alcoholysis reactions typically employ activated esters. As this work utilizes in situ-generated oxazolone intermediates, could the authors clarify whether pre-activated amino acid esters (e.g., p-nitrophenyl or pentafluorophenyl esters) were tested as substrates?

Our response: We appreciate the reviewer's valuable suggestion. Given that oxazolone intermediates suitable for our protocol are available, we investigated the use of pre-activated p-nitrophenyl amino acid esters as substrates for this purpose. We were pleased

to find that the desired chiral amino esters could be obtained in high yield with excellent enantioselectivity, as demonstrated below and listed in **Table S5** of the Supplementary Information.

This result confirms that our NHC-catalyzed DKR methodology is indeed compatible with pre-activated amino acid esters, providing an alternative synthetic route that bypasses the in situ formation of oxazolones. The comparable stereoselectivity achieved with *p*-nitrophenyl esters further validates the robustness of our catalytic system. It suggests that the stereocontrol mechanism remains consistent regardless of whether the activation occurs in situ or through pre-activated substrates.

2. DIC was identified as the optimal activating reagent, while EDCI yielded inferior stereoselectivity despite structural similarity. What mechanistic rationale (e.g., steric, electronic, or kinetic factors) might explain this disparity in stereocontrol?

Our response: We thank Reviewer 1 for the insightful questions and constructive feedback. Although DIC and EDCI share structural similarities, several key differences become particularly pronounced under our reaction conditions. Compared to EDCI, DIC exhibits higher reactivity in organic solvents and forms *O*-acylisourea intermediates that are more stable under non-aqueous conditions.

Our DFT mechanistic calculations reveal that oxazolones serve as crucial intermediates for the reaction progression and play a key role in NHC-catalyzed stereocontrol. Mechanistically, certain *O*-acylisourea intermediates can cyclize to form oxazolones (*Chem. Rev.* **2011**, *111*, 6557–6602). We propose that when using EDCI, the rate of in situ oxazolone formation may be slower than the competing alcoholysis rate in the presence

of the alcohol component and the strong base Cs_2CO_3 . This kinetic imbalance leads to direct formation of the amino ester product via alcoholysis, resulting in poor stereocontrol as the stereodetermining oxazolone-mediated pathway is bypassed.

Additionally, there is another possible pathway in which the more highly activated O-acylisourea intermediate generated from EDCI can directly react with the free NHC catalyst to form the acylazolium intermediate, thereby circumventing the oxazolone formation step. This direct pathway lacks the crucial chiral matching process between the oxazolone and the NHC catalyst. Consequently, the acylazolium intermediate formed via this direct route may exhibit lower stereoselectivity compared to the pathway involving oxazolone participation, where optimal chiral recognition and stereocontrol are achieved through the oxazolone-NHC interaction.

3. For Ts- and Cbz-protected substrates in Figure 5b, stereoselectivity was observed despite no oxazolone intermediate formation. Could this imply a transient active ester pathway (via DIC-activated amino acids) that generates acyl azolium intermediates directly upon NHC interaction, thereby enabling stereocontrol?

Our response: We appreciate the reviewer's thoughtful comment. As discussed above, we agree that there is indeed a possibility of a transient active ester pathway where O-acylisourea intermediates generated via DIC-activated amino acids can be directly captured by the free NHC catalyst to generate acyl azolium intermediates, thereby enabling stereocontrol.

The results shown in **Figure 5b** support this hypothesis, where Ts- and Cbz-protected substrates exhibit high yields but lower enantioselectivities compared to the oxazolone-mediated pathway. This suggests that while the direct O-acylisourea-NHC pathway could provide some degree of stereocontrol, it is inherently less selective than the oxazolone-mediated route. The moderate enantioselectivities observed indicate that acyl azolium intermediates formed through this direct pathway retain some chiral information. However, the stereodetermining step may be less well-defined compared to the highly organized oxazolone-NHC interaction. This dual mechanistic scenario explains why certain protecting groups (Ts, Cbz) that disfavor oxazolone formation can still deliver enantioenriched products, albeit with reduced selectivity.

4. To enhance mechanistic clarity, including 3D structural data for all intermediates and transition states in the Supporting Information is recommended. Only selected structures (Figure 7, main text) are currently provided.

Our response: We sincerely thank Reviewer 1 for this constructive suggestion. All optimized geometries, along with detailed structural parameters for intermediates and

transition states, are presented in **Figure S5** of the Supplementary Information.

5. SI Table 4 reports limited diastereoselectivity (72% de) for amino acid ester substrates in chiral peptide synthesis. Were efforts made to optimize this transformation (e.g., activating reagents, catalysts, solvents, temperatures)? Additionally, is compatibility with sulfonamide-based reagents feasible?

Our response: We appreciate the reviewer's valuable suggestion. In our initial study, we have already screened various reaction parameters, including different activating reagents, NHC catalysts, solvents, and temperatures to optimize the diastereoselectivity for amino acid ester substrates in chiral peptide synthesis. While we achieved 72% de under the current optimized conditions, we acknowledge that further improvement is desirable for practical applications.

Regarding compatibility with sulfonamide-based reagents, we have explored this avenue during our optimization studies. Unfortunately, when sulfonamide-based coupling reagents were employed, the reactions proceeded with complete loss of stereoselectivity, yielding racemic products:

6. Manuscript Corrections:

a) A discrepancy exists in the discussion of Int2 conformers: The cited 2.9 kcal/mol energy difference (Figure 7) conflicts with the textual description. Clarification is required.

Our response: We thank Reviewer 1 for bringing this inconsistency to our attention. We have carefully checked and corrected it in the manuscript.

b) In Figure 6a (heatmap), the yield and es values for the Ac-protected substrate under condition B appear mislabeled. Please verify and correct.

Our response: We apologize for the negligence. We have corrected the yield and es values for the Ac-protected substrate under condition B in **Figure 6a**. The figure has been updated to reflect the accurate experimental data.

Reviewer #2:

This manuscript describes the stereoselective diversification of α -amino acids using N-heterocyclic carbene catalysts in the presence of a Cs_2CO_3 base. The resulting peptides exhibit high enantio- and diastereoselectivities. The synthetic methodology presented is significant for applications in organic synthesis and medicinal chemistry. They also used

DFT calculations to gain insight into the mechanisms, and origin of the stereoselectivities. As a reviewer with expertise in DFT calculations, I would like to offer several comments regarding the computational aspects of the paper.

Our response: We appreciate the reviewer's supportive feedback and have addressed the points raised, as detailed below.

Comments:

1. In Figures 3–5, the authors use the term "enantioexcess (ee)," whereas elsewhere they refer to "enantioselectivity (es)." Although these terms are related, the inconsistency may confuse readers. The authors are encouraged to standardize the terminology throughout the manuscript.

Our response: We thank the reviewer for drawing attention to this terminology. In our manuscript, we have intentionally employed two distinct terms to reflect the different substrate types and mechanistic contexts:

- For reactions conducted on racemic or achiral substrates, we report the enantiomeric purity of the isolated product as "enantioexcess (ee)."
- For kinetic resolutions or transformations involving enantiomerically enriched starting materials, we use "enantioselectivity (es)" to emphasize the preferential conversion of one enantiomer over the other.

This usage aligns with conventions in the asymmetric catalysis literature, where "ee" denotes the enantiomeric purity of products derived from racemic or achiral precursors, and "es" (sometimes referred to as the selectivity factor) describes the differential reaction rates in kinetic resolutions of chiral substrates.

2. Dispersion Correction Clarification: In the caption of Figure 7, the authors mention the use of D3 dispersion corrections with Becke–Johnson (BJ) damping. However, the Supplementary Information does not specify whether BJ damping was applied (maybe zero damping?). The authors should clarify whether B3LYP-D3(BJ) or B3LYP-D3 was used in their calculations.

Our response: We appreciate Reviewer 2's careful attention to this technical detail. We confirm that B3LYP-D3(BJ) was used in our calculations. We have clarified this in the Computational Methods section of the Supplementary Information, which now reads: "Geometry optimizations were carried out at B3LYP functional with dispersion energy corrections by Grimme's dispersion correction D3BJ and def2-SVP basis set for all atoms."

3. Mechanistic Insight: Figures 7 and S1 indicate that the (S)-pathway is favored over the (R)-pathway. However, the explanation focuses solely on steric repulsions in the acyl

azolium intermediate (Figure S3). The authors are encouraged to perform energy decomposition analyses (EDA) and noncovalent interaction (NCI) analyses on the transition states to deepen the mechanistic understanding. These approaches could provide more detailed insights into the origin of stereoselectivity.

Our response: We thank Reviewer 2 for this insightful suggestion to deepen our mechanistic understanding. Following the recommendation, we have performed IGMH analysis to investigate the non-covalent interactions between acyl azolium, alcohol, and base fragments in both (*R*)-**TS₃** and (*S*)-**TS₃**. As shown in **Figure S4**, our analysis reveals that (*S*)-**TS₃** is stabilized by additional π - π interactions (highlighted by purple circles). Together with the reduced steric hindrance in the (*S*)-configured acyl azolium intermediate, these two factors collectively determine the stereoselectivity of the reaction.

Additional Comments:

4. In Figure S1, "Ts" should be corrected to "TS" to maintain consistency with standard nomenclature.

Our response: We apologize for the negligence. We have corrected "Ts" to "TS" in **Figure S1-2** and have carefully reviewed the entire manuscript to ensure consistent use of standard nomenclature throughout.

5. In the computational methods section of the Supplementary Information, the authors should specify how Gibbs free energies were calculated, including the temperature and pressure conditions used.

Our response: We sincerely thank Reviewer 2 for his/her professional suggestion. We have added the following description in the computational methods section: "Gibbs free energies were calculated under 298.15K and 1 atm pressure."

Reviewer #3:

In the manuscript by Jean Chen, Yong Huang et al., an N-heterocyclic carbene (NHC)-

catalyzed strategy enabling enantioselective synthesis of α -(U)AA esters and peptides is reported. Stereodivergent synthesis of both enantiomers of chiral α -Aas derivatives was accomplished with >90% ee (de) via the DIC-initiated formation of racemic azlactones followed by efficient deracemization by means of NHC-generated acyl azolium intermediates and proton shuttling. The reaction mechanism was supported by control experiments and DFT calculations. The method is applicable to diverse substrates and demonstrates a broad functional group tolerance. It may be useful as a potential platform for asymmetric synthesis of peptide medications.

However, manuscript has some drawbacks which require major corrections.

Our response: We sincerely thank Reviewer 3 for his/her insightful questions and constructive feedback. We have addressed the points raised, as detailed below.

1. There is no data on scalability of the proposed NHC-based deracemization procedure which is crucial for biomolecular design and the development of peptide therapeutics.

Our response: We thank the reviewer for this comment regarding the scalability data. We have demonstrated the scalability of our NHC-based deracemization procedure through a gram-scale reaction as shown in **Scheme S8** of the Supplementary Information. Additionally, we have expanded the substrate scope with several new examples, which are presented in **Table S8** of the Supplementary Information. These results collectively demonstrate the robustness and broad applicability of our methodology.

2. Experimental details of the declared Wang resin-based solid-phase peptide synthesis (SPPS) protocol should be added and discussed.

Our response: We appreciate the reviewer's suggestion. We have added detailed experimental procedures for the Wang resin-based SPPS protocol in the Supplementary Information (**Scheme S12**). An introductory information to these protocols has been appended to the legend of **Figure 6** ([f] For concrete operations, see **Schemes S12**). The added procedures include comprehensive details for resin loading, amino acid coupling, deprotection cycles, and cleavage conditions used in the synthesis of peptide **8c**.

3. The CCDC number «2223535» provided by the authors in the manuscript (Figure 3) and the ESI (Section «determination of absolute configuration», Page 82) must be checked and corrected. Quite different compound with the same CCDC number has been recently reported by another research team (Zi-Chao Wang, Xiaohua Luo, Jia-Wen Zhang, Chen-Fei Liu, Ming Joo Koh, Shi-Liang Shi, Nature Catalysis, 2023, 6, 1087, DOI: 10.1038/s41929-023-01037-9).

Our response: We thank the reviewer for bringing this critical issue to our attention. We

apologize for this oversight and appreciate the reviewer's diligence in identifying this error, which helps ensure the accuracy and integrity of our crystallographic data. We have updated our crystal structure data (**4ai**, CCDC: 2469893; **4al**, CCDC: 2469924) in both the manuscript (**Figure 3**) and the Supplementary Information (Section "determination of absolute configuration", Page 82).

4. One question more. The authors determined the absolute (S)-configuration of product 4e basing on the X-ray crystal structure. However, the determined space group «P -1» is centrosymmetric in which only racemic or achiral substances can crystallized. It is not clear, how the authors could determine the absolute configuration using the P-1 crystal.

Our response: We thank the reviewer for raising this critical crystallographic concern. We acknowledge that there appears to be an inconsistency in our reported data. The reviewer is correct that space group P-1 is centrosymmetric and cannot be used to determine absolute configuration, as it can only accommodate racemic or achiral compounds. We apologize for the confusion in our crystallographic data reporting. We have currently updated our crystal structure determination in both the main text (**Figure 3**) and the Supplementary Information (Section "determination of absolute configuration", Page 82).

5. Melting points of solid products 4, 6 and 8 are absent in the ESI.

Our response: We thank Reviewer 3 for the suggestion. To increase the data diversity, we have supplied the melting points of solid products in the Supplementary Information.

6. Structural formulas of NHC-3s and NHC-3c mentioned in captions to Figures 3 and 4 should be depicted not only in the SI but also in the manuscript.

Our response: We thank the reviewer for the professional suggestion. To enhance clarity for readers, we have included the structural formulas of NHC-3s and NHC-**3c** in **Figures 3** and **4** of the manuscript, ensuring that the discussion flows more logically and is easier to follow.

7. I recommend the authors to measure yield, ee and dr values in percents in rectangles embedded to Figures 5b and 6a.

Our response: We appreciate the reviewer's suggestion. We modified the expression format for the yield, ee, and dr values in the rectangles embedded in Figures 5b and 6a of the revised manuscript as suggested.

8. Figure 5c. The hydroxyproline ester 4aj can't be obtained under standard condition A including usage of nucleophile 2a.

Our response: We thank the reviewer for bringing this point to our attention. We apologize for not clearly expressing the specific nucleophile we used. To obtain the chiral hydroxyproline ester **4aj**, we used N-fmoc-trans-4-hydroxy-L-proline methyl ester as a nucleophile rather than **2a**. We modified the sentence in the revised manuscript and added the specific nucleophile structure format in **Figure 5c** to facilitate the reader's understanding.

Page 6, line 150: "This stereochemical control was further validated using (rac)-**1aa** as the substrate and N-fmoc-trans-4-hydroxy-L-proline methyl ester as the nucleophile, which allowed for the synthesis of a single stereoisomer (*S*) or (*R*) **4aj**."

9. Page 7, line 166: The 99% es – is it a "moderate" stereoselectivity?

Our response: We thank the reviewer for their careful reading of the manuscript. We have modified the expression in the manuscript: "The reaction efficiently operated under base-free conditions, outperforming direct coupling approaches that exhibited moderate yield and excellent stereoselectivity (73% yield, 99% es vs. entries 1-3)"

10. Pages in the ESI should be numerated.

Our response: We appreciate the reviewer's reminder. Page numbers have been appended.

Reviewer #4:

Our response: We sincerely appreciate the time and effort the reviewer has dedicated to evaluating our manuscript.

Reviewer's Comments to Author (Responses are highlighted in red):

Reviewer #3 :

1) The authors revised the original manuscript thoroughly according to the review comments. However, one more question arose after the revision. In Supplementary Information, page 85 there is no scheme 12, which is expected illustrate solid phase synthesis of compound **8c** in detail. Step 1 also raises questions as it describes synthesis of “desired product” **1ab** from the same compound **1ab**. The necessity of this step and a role of the Wang resin in this “transformation” is not clear.

I recommend publication of this manuscript after the authors address this comment.

Our response: Thank you for your careful review and comments. We apologize for any confusion that may have caused. The detailed synthesis protocol for compound **8c** is shown in **Figure S12**, which appears on page 85 of the Supplementary Information. Regarding Step 1 and the role of Wang resin: In this step, we perform a chiral resolution, using Wang resin as a solid support to convert racemic compound **1ab** into the desired chiral (*S*)-**1ab**. Wang resin not only acts as a solid support but also helps convert the racemic substrate into the chiral product by condensing its hydroxyl group with the racemic carboxylic acid under the influence of NHC catalyst, thereby enabling in situ chiral amino acid synthesis. We hope this serves as a starting point for applying this methodology to solid-phase synthesis. We hope these clarifications address your concerns. We will revise the Supplementary Information to clarify these points in the final version.